# The Thermomajorization Polytope and Its Degeneracies

**DOI:** 10.3390/e26020106

**Published:** 2024-01-24

**Authors:** Frederik vom Ende, Emanuel Malvetti

**Affiliations:** 1Dahlem Center for Complex Quantum Systems, Freie Universität Berlin, 14195 Berlin, Germany; 2School of Natural Sciences, Technische Universität München, 85737 Garching, Germany; emanuel.malvetti@tum.de; 3Munich Center for Quantum Science and Technology (MCQST) & Munich Quantum Valley (MQV), 80799 München, Germany

**Keywords:** quantum thermodynamics, thermal operations, thermomajorization, thermal equilibrium, Gibbs-stochastic matrix, cyclic process

## Abstract

Drawing inspiration from transportation theory, in this work, we introduce the notions of “well-structured” and “stable” Gibbs states and we investigate their implications for quantum thermodynamics and its resource theory approach via thermal operations. It is found that, in the quasi-classical realm, global cyclic state transfers are impossible if and only if the Gibbs state is stable. Moreover, using a geometric approach by studying the so-called thermomajorization polytope, we prove that any subspace in equilibrium can be brought out of equilibrium via thermal operations. Interestingly, the case of some subsystem being in equilibrium can be witnessed via the degenerate extreme points of the thermomajorization polytope, assuming that the Gibbs state of the system is well structured. These physical considerations are complemented by simple new constructions for the polytope’s extreme points, as well as for an important class of extremal Gibbs-stochastic matrices.

## 1. Introduction

The core idea of quantum thermodynamics—a field that has gained significant traction within the last decade—is to apply thermodynamic ideas to (ensembles of) systems of finite size, instead of “thermodynamically large” setups [1]. While this discipline comes with a number of principal questions and concepts (such as fluctuation theorems, thermal machines, the fundamental laws, thermalization, and many more; cf. [2]), an all-round successful approach has been to take the open systems perspective [3]. There, one models changes to a system of interest as the reduced action of a larger, closed system (i.e., system plus some form of environment, such as a bath) through some total Hamiltonian $H_{\mathrm{tot}}=H_{S}+H_{B}+H_{I}$ consisting of a system, an environment, and an interaction term. (To avoid any confusion, let us point out explicitly that the individual Hamiltonians $H_{S},H_{B}$ are to be understood as local terms $H_{S}⊗1_{E},1_{S}⊗H_{B}$, in particular, when adding them). Typical thermodynamic constraints imposed on top are that the environment (bath) starts out in equilibrium, i.e., in a Gibbs state, or that the total energy is conserved, i.e., $\left[H_{\mathrm{tot}},H_{S}+H_{B}\right]=\left[H_{I},H_{S}+H_{B}\right]=0$. Note that some works have weakened energy conservation to average energy conservation in the past. This leads to a framework where the quantum free energy characterizes all possible state changes [4].

This perspective ties into the information-theoretic and, more specifically, into the resource theory approach to quantum thermodynamics. There, one attempts to formalize which operations can be carried out at no cost (with respect to some resource, e.g., work), and one of the main aspects of this approach is to characterize when state transfers under thermodynamic constraints are possible. This comes with a subset of quantum channels called *thermal operations*, which can be carried out without having to expend any resource. They arise from the above open systems construction as follows [5,6,7]. Given an *n*-level system described by some system’s Hamiltonian $H_{S}\in iu\left(n\right)$, as well as some fixed background temperature T>0 that every bath needs to adopt, the set $\mathrm{TO}(H_{S},T)$ of all thermal operations with respect to $H_{S},T$ is defined as
$$\begin{array}{c} \left\{{\mathrm{tr}}_{B}\left(e^{iH_{\mathrm{tot}}}\left(\left(\cdot \right)⊗\frac{e^{-H_{B}/T}}{\mathrm{tr}(e^{-H_{B}/T})}\right)e^{-iH_{\mathrm{tot}}}\right):\;\begin{array}{c} m\in N,H_{B}\in iu\left(m\right),\;H_{\mathrm{tot}}\in iu\left(mn\right)\\ \left[H_{\mathrm{tot}},H_{S}+H_{B}\right]=0 \end{array}\right\}\;. \end{array}$$
Here, u(m) is the unitary Lie algebra in *m* dimensions so iu(m) is the collection of Hermitian m×m matrices. As hinted at before, one of the central questions of this framework is the following: given an initial and a target state of some quantum system, can the former be transformed into the latter by means of a thermal operation?

The resource theory approach to quantum thermodynamics leads to a number of structural insights, ranging from optimal protocols for work extraction [4] and cooling [8] to the so-called second laws of quantum thermodynamics [9,10]. The latter precisely relates to the above state interconversion problem via a generalization of classical majorization called “thermomajorization”. First described by Ruch et al. [11] in the 1970s, thermomajorization has gained widespread popularity over the last decade due to the influential works of Brandão et al. [10], Horodecki and Oppenheim [9], and Renes [12], as well as many more [8,13,14,15,16,17]; there, among others, it has been used to solve the state interconversion problem in the quasi-classical realm; more details on this are given below. However, one can tackle this problem from another, more geometric perspective: for this, one abstractly defines the collection of all states that can (approximately) be generated via thermal operations starting from some initial state ρ
$$M_{H_{S},T}\left(\rho \right):=\left\{\Phi \left(\rho \right):\Phi \in \bar{\mathrm{TO}(H_{S},T)}\right\}\;,$$
and then one studies the geometric properties of $M_{H_{S},T}\left(\rho \right)$. This set has been called the *(future) thermal cone* (Note that the term “cone” originates from light cones in general relativity and should not be confused with (convex) cones from linear algebra) [18,19,20] or, in the case of quasi-classical states ρ, the *thermal polytope* [8], and the elements of $M_{H_{S},T}\left(\rho \right)$ are precisely those states that are said to be *thermomajorized* by ρ. In the quasi-classical case $[\rho ,H_{S}]=0$, the structure of the thermal cone is known to simplify considerably. This is due to the following crucial facts:Every thermal operation leaves the set of quasi-classical states invariant [21]: if $[\rho ,H_{S}]=0$, then $[\Phi \left(\rho \right),H_{S}]=0$ for all $\Phi \in \bar{\mathrm{TO}(H_{S},T)}$.Thermal operations and general Gibbs-preserving quantum channels (approximately) coincide on quasi-classical states when considering only the diagonal (Section 3.4 [22]).

Combining these facts shows that for any state ρ with $[\rho ,H_{S}]=0$ where $H_{S}$ is non-degenerate (i.e., ρ is diagonal in “the” eigenbasis of $H_{S}$), $M_{H_{S},T}\left(\rho \right)$ equals the set of all diagonal states Φ(ρ) where Φ is a quantum channel that preserves the Gibbs state $e^{-H_{S}/T}/Z$. However, for diagonal states—where we write $x,y\in R^{n}$ for the vectors of their respective diagonal entries—the existence of such a channel is equivalent to the existence of a Gibbs-stochastic matrix *A* such that Ay=x (Corollary 4.5 [23]). For this recall that given some Hamiltonian $H_{S}=\mathrm{diag}\left(E_{1},\ldots ,E_{n}\right)$, as well as T>0, a matrix *A* is called Gibbs-stochastic if the entries of *A* are non-negative, every column of *A* sums up to one, and Ad=d. Here, $d:={\left(e^{-E_{j}/T}\right)}_{j=1}^{n}$ is the (unnormalized, but can also be normalized) vector of Gibbs weights. Moreover, we write $s_{d}\left(n\right)$ for the collection of all Gibbs-stochastic matrices with regard to the Gibbs vector *d*. Thus, $M_{H_{S},T}\left(\mathrm{diag}\left(y\right)\right)$ equals $\left\{\mathrm{diag}\left(Ay\right):A\in R^{n\times n}\;\mathrm{Gibbs}-\mathrm{stochastic}\right\}$ where one, instead and equivalently, can focus solely on the diagonal by considering the so-called *thermomajorization polytope*
(1)$$M_{d}\left(y\right):=\left\{Ay:A\in R^{n\times n}\;\mathrm{Gibbs}-\mathrm{stochastic}\;\right\}\;.$$
At first glance, focusing on the quasi-classical case may appear fruitless in understanding the behavior of quantum systems. However, we point out that optimal cooling protocols rely on two-level Gibbs-stochastic matrices [8], which can be realized within the Jaynes–Cummings model [15]. Moreover, taking the quasi-classical perspective allows one to identify thermal operations that are simple to implement experimentally [24], and tools from the quasi-classical case have been used in quantum control theory to find non-trivial upper bounds on reachable states for dissipative bath couplings [25,26].

In this work, we will combine this geometric approach to quantum thermodynamics with the established field of transportation theory. While this is not the first time that these fields are being linked—cf. Section 2.2 for a short review and Section 3.2 for some results that this perspective has lead to already—this paper’s main idea is not to take tools, but rather (not as obvious, yet key) concepts from transportation theory and to investigate their implications in quantum thermodynamics. Thus, this work is structured as follows. We begin by reviewing characterizations of thermomajorization in the quasi-classical realm in Section 2.1, followed by a review of the basics of transportation theory in Section 2.2. Therein, we also introduce (or rather, translate) the key concepts of “stable” and “well-structured” Gibbs states, which will emerge to be quite intuitive. The implications of these notions will be the topic of Section 3, which contains the main results. More precisely, stable Gibbs states are found to be in one-to-one correspondence with the impossibility of (global) cyclic state transfers—which will also lead us to the notion of a “subspace in equilibrium”—cf. Section 3.1. The latter notion is found to be closely connected to the geometry of $M_{d}\left(y\right)$ and is reflected in the (number of) extreme points of $M_{d}\left(y\right)$, assuming that the Gibbs state is well structured (Section 3.2). The point of view taken in this work also leads to simple, intuitive ways to construct extreme points, as well as Gibbs-stochastic matrices that realize the corresponding state transfer; this is what the example and visualization Section 4 will illustrate.

## 2. Preliminaries and Review

### 2.1. Thermomajorization for Quasi-Classical States

The object related to thermomajorization most commonly found in the literature is the following: let any $d\in R_{++}^{n}$ (that is, $d\in R^{n}$ with d>0 as *d* plays the role of the vector of Gibbs weights) as well as $y\in R^{n}$ be given. One defines the *thermomajorization curve of y (with respect to d)*, denoted by ${\mathrm{th}}_{d,y}$, as the piecewise linear, continuous curve fully characterized by the elbow points ${\left\{\left(\sum_{i=1}^{j}d_{\tau (i)},\sum_{i=1}^{j}y_{\tau (i)}\right)\right\}}_{j=0}^{n}\;$, where $\tau \in S_{n}$ is any permutation such that $\frac{y_{\tau (1)}}{d_{\tau (1)}}\geq \ldots \geq \frac{y_{\tau (n)}}{d_{\tau (n)}}$. Equivalently (Remark 7 [27]), ${\mathrm{th}}_{d,y}:\left[0,e^{⊤}d\right]\to R$ for all $c\in [0,e^{⊤}d]$ satisfies
(2)$$\begin{array}{cc} {\mathrm{th}}_{d,y}\left(c\right) & =\min_{\left\{i=1,\ldots ,n\;:\;d_{i}>0\right\}}\left(\left(\sum_{j=1}^{n}\max\left\{y_{j}-\frac{y_{i}}{d_{i}}d_{j},0\right\}\right)+\frac{y_{i}}{d_{i}}c\right)\\ \; & =\frac{e^{⊤}y}{2}+\min_{\left\{i=1,\ldots ,n\;:\;d_{i}>0\right\}}\left(∥y-\frac{y_{i}}{d_{i}}d{∥}_{1}+\frac{y_{i}}{d_{i}}\left(c-\frac{e^{⊤}d}{2}\right)\right) \end{array}$$
where, here and henceforth, $e:={\left(1,\ldots ,1\right)}^{⊤}$ and ${∥\cdot ∥}_{1}$ is the usual vector 1-norm. Note that this curve is a generalization of the notion of Lorenz curves from majorization theory (p. 5 [28]). While the condition $d_{i}>0$ from the minimum in (Equation 2) is redundant for now, it allows us to generalize the definition of ${\mathrm{th}}_{d,y}$ to the zero-temperature case where some of the entries of *d* vanish; cf. Remark 2 below.

**Remark** **1.**
*It is clear from the definition that the thermomajorization curve is invariant under permutations in the sense that ${\mathrm{th}}_{\underline{\sigma }d,\underline{\sigma }y}\equiv {\mathrm{th}}_{d,y}$ for all $\sigma \in S_{n}$. Here, given some permutation $\sigma \in S_{n}$, we write $\underline{\sigma }$ for the corresponding permutation matrix $\sum_{i=1}^{n}e_{i}e_{\sigma (i)}^{⊤}$. In particular, the identities $\underline{\sigma \circ \tau }=\underline{\tau }\cdot \underline{\sigma }$, ${\left(\underline{\sigma }x\right)}_{j}=x_{\sigma (j)}$, and ${\left(\underline{\sigma }A\underline{\tau }\right)}_{jk}=A_{\sigma \left(j\right)\tau^{-1}\left(k\right)}$ hold for all $A\in R^{n\times n}$, $x\in R^{n}$, j,k=1,…,n, and all $\sigma ,\tau \in S_{n}$.*


Now, the precise connection between thermomajorization and (quasi-classical) state transfers is summarized in the following well-known result. Given any $x,y\in R^{n}$, the following statements are equivalent (Proposition 1 [27]).

There exists a Gibbs-stochastic matrix *A* such that Ay=x. We denote this by $x{≺}_{d}y$.$e^{⊤}x=e^{⊤}y$ and ${\mathrm{th}}_{d,x}\left(c\right)\leq {\mathrm{th}}_{d,y}\left(c\right)$ for all $c\in [0,e^{⊤}d]$.$e^{⊤}x=e^{⊤}y$ and ${\mathrm{th}}_{d,x}\left(\sum_{i=1}^{j}d_{\tau (i)}\right)\leq {\mathrm{th}}_{d,y}\left(\sum_{i=1}^{j}d_{\tau (i)}\right)$, i.e.,
$$\sum_{i=1}^{j}x_{\tau (i)}\leq \min_{\left\{i=1,\ldots ,n\;:\;d_{i}>0\right\}}\left(\left(\sum_{j=1}^{n}\max\left\{y_{j}-\frac{y_{i}}{d_{i}}d_{j},0\right\}\right)+\frac{y_{i}}{d_{i}}\left(\sum_{k=1}^{j}d_{\tau (k)}\right)\right)$$
for all j=1,…,n−1, where $\tau \in S_{n}$ is any permutation such that $\frac{x_{\tau (1)}}{d_{\tau (1)}}\geq \ldots \geq \frac{x_{\tau (n)}}{d_{\tau (n)}}$.${∥x-td∥}_{1}\leq {∥y-td∥}_{1}$ for all t∈R.$e^{⊤}x=e^{⊤}y$ and $∥d_{i}x-y_{i}{d∥}_{1}\leq {∥d_{i}y-y_{i}d∥}_{1}$ for all i=1,…,n.

These criteria slightly simplify for probability vectors $x,y\in R^{n}$ (e.g., containing the spectrum of any two quantum states), i.e., for vectors x,y≥0 such that $e^{⊤}x=e^{⊤}y=1$: in this case, the “thermomajorization curve” criterion reduces to ${\mathrm{th}}_{d,x}\left(c\right)\leq {\mathrm{th}}_{d,y}\left(c\right)$ for all $c\in [0,e^{⊤}d]$ (resp. all $c\in \{\sum_{i=1}^{j}d_{\tau (i)}:j=1,\ldots ,n-1\}$, where τ sorts $\frac{x}{d}$ non-increasingly). Starting from the thermomajorization curves, there even exists an algorithm to find a Gibbs-stochastic matrix that implements the state transition in question [22]. Nonetheless, the reason that thermomajorization curves are equivalent to conditions using the 1-norm is their fundamental link by means of the Legendre transformation; recall that, given a convex function f:D→R on a connected domain D⊆R, its Legendre–Fenchel transformation $f^{*}:D^{*}\to R$ is defined to be $f^{*}\left(p\right):=\sup_{x\in D}\left(px-f(x)\right)$ for all $p\in D^{*}:=\left\{p\in R\;:\;\sup_{x\in D}\left(px-f\left(x\right)\right)<\infty \right\}$ (Chapter 3.3 [29]). Indeed, for all non-zero d≥0, all $y\in R^{n}$, and all t∈R, one has
(3)$$2{\left(-{\mathrm{th}}_{d,y}\right)}^{*}\left(t\right)=e^{⊤}y+t\left(e^{⊤}d\right)+{∥y+td∥}_{1}$$
This readily follows from the definition of the Legendre transformation together with the fact that ${\mathrm{th}}_{d,y+td}\left(c\right)={\mathrm{th}}_{d,y}\left(c\right)+ct$ for all $c\in [0,e^{⊤}d]$, as well as the fact that the maximum of any thermomajorization curve equals the sum over all non-negative entries of the initial state (Lemma 13 (iii) [27]). With this, the equivalence of the corresponding characterizations of *d*-majorization is due to the fact that the Legendre transformation in an involution that respects (more precisely, reverses) the order (Theorem 4.2.1 [29]). While these conditions are mathematically equivalent, in practice, it is often easier to prove the results using the curves ${\mathrm{th}}_{d,y}$. This empirical observation will also be supported by the main part of this paper.

**Remark** **2**(Thermomajorization for Zero Temperature). *The above conditions for thermomajorization remain well defined even if some entries of d vanish, assuming that ${\mathrm{th}}_{d,y}$ is defined via (Equation 2). Physically, this relates to the temperature being zero, in which case all entries of d that do not correspond to the zero-point energy of the system vanish. If this lowest energy level with corresponding index j is non-degenerate, then $d=d\left(T\right)\to e_{j}$ as $T\to 0^{+}$. Interestingly, the above characterization of thermomajorization remains valid in this regime. For proof, we refer to Appendix A. This explains the known continuity problems of the associated polytope (cf. Example 3 in [27]). If some of the $d_{i}$ vanish, so do the corresponding inequalities, meaning that there are less restrictions. Hence, the polytope can only become larger in this case.*

The importance of the 1-norm conditions is that they lead to a remarkable characterization of $M_{d}\left(y\right)$ from (Equation 1); cf. (Theorem 10 [27]):(4)$$M_{d}\left(y\right)=\left\{x\in R^{n}\;:\;e^{⊤}x=e^{⊤}y\;\mathrm{and}\;\forall_{m\in {\left\{0,1\right\}}^{n}}\;m^{⊤}x\leq {\mathrm{th}}_{d,y}\left(m^{⊤}d\right)\right\}$$
The geometric interpretation of each of these inequalities is that every binary vector $m\in {\left\{0,1\right\}}^{n}\setminus \left\{0,e\right\}$ is the normal vector to a halfspace that limits $M_{d}\left(y\right)$. The location of said halfspace is determined by the value ${\mathrm{th}}_{d,y}\left(m^{⊤}d\right)$ and thus by the thermomajorization curve. Note that the orientation of these halfspaces is universal, i.e., they are independent of any of the system parameters; subsequently, y,d only influence the *location* of the faces. Another means of expressing this is to say that thermomajorization polytopes are obtained by shifting the faces of a classical majorization polytope. Note that this can lead to the situation where some of these halfspace conditions become redundant. This description of $M_{d}\left(y\right)$ has been used to prove continuity of the map $\left(d,P\right)\mapsto M_{d}\left(P\right)$, where d>0 and *P* is from the collection of non-empty compact sets in $R^{n}$ equipped with the Hausdorff metric (Theorem 12 (ii) [27]). Alternatively, this result can be obtained from the continuity of the set of Gibbs-stochastic matrices in $H_{0}$ and T>0 (Theorem 5.1 [30]).

Now, writing the bounded set $M_{d}\left(y\right)$ as the solution to finitely many inequalities shows that it is a convex polytope, i.e., it ultimately can be written as the convex hull of finitely many points [31]. Rather than merely being an abstract result, these extreme points have an analytic form. Moreover, halfspaces becoming redundant leads to the coalescence of extreme points; more details on this are given in Section 3.2.

### 2.2. Transportation Polytopes

It emerges that Gibbs-stochastic matrices are closely related to transportation matrices, which are at the core of the well-studied field of transportation theory [32,33,34,35]. It appears that, so far, this has been very much overlooked: this notion does not even appear in the standard work on majorization by Marshall et al. [28], and the only papers from the quantum thermodynamics literature (that we are aware of) that have used results from transportation theory are ones by Mazurek et al. [17,36].

Assuming finite domains, *transportation matrices* are non-negative matrices with fixed column and row sums. More precisely, these are matrices $A\in R_{+}^{m\times n}$ such that Ae=r and $c^{⊤}=e^{⊤}A$ for some $r\in R^{m}$, $c\in R^{n}$ with $e^{⊤}r=e^{⊤}c$. The collection of all such matrices is called the *transportation polytope* and is often denoted by T(r,c). As already observed by Hartfiel [30], the connection to our setting then is the following. For non-zero temperatures (i.e., d>0), there is a one-to-one correspondence between (the extreme points of) the Gibbs-stochastic matrices and (the extreme points of) the symmetric transportation polytope T(d,d) by means of the isomorphism X↦Xdiag(d). From our point of view, pursuing this approach may seem counter-intuitive because the geometry of the Gibbs-stochastic matrices is known to be more complicated than the thermomajorization polytope. Already, in three dimensions, the number of extreme points of the Gibbs-stochastic matrices depends on the temperature of the bath (Figure 1 [36]), i.e., on certain relations between the entries of *d* (Appendix A [27]). However, drawing this connection grants access to powerful tools from combinatorics and graph theory. Roughly speaking, there is a relation between the extreme points of T(d,d) and the spanning trees of the associated bipartite graphs. We do not provide details of the underlying techniques (instead, we refer to [17]); rather, we adopt useful notions from this field and adapt them to thermomajorization as well as the associated polytope. For this, our starting point is a paper by Loewy et al. [37], where conditions on the vector *d* that classify certain features of (the polytope of) Gibbs-stochastic matrices are identified. We present these conditions—which Loewy et al. simply call “property (a)” and “property (b)” (To be precise, while Definition 1 (i) is the same as their “property (a)”, Definition 1 (ii) strengthens “property (b)”. However, (i) and (ii) together are equivalent to property (a) and property (b) combined.)—in the following definition.

**Definition** **1.**
*Given $d\in R_{++}^{n}$, define a map D:P({1,…,n})→[0,∞) on the power set of {1,…,n} via $D\left(I\right):=\sum_{i\in I}d_{i}$.*


*(i)* *We say that d is* well structured *if, for all I,J⊆{1,…,n}, |I|<|J|, one has D(I)<D(J).**(ii)* *We call d* stable *if D is injective.*
Some remarks on these notions are in order. By definition, *d* is stable if summing up two sets of entries of *d* only yields the same result if the entries coincide in the first place. In particular, stability implies that *d* is non-degenerate. Note that, for non-degenerate systems, stability is a generic property as only finitely many temperatures give rise to unstable Gibbs states. On the other hand, *d* is well structured if summing up k−1 arbitrary entries of *d* always yields less than summing up any *k* entries of *d*. Interestingly, this notion is fully captured by the inequality
(5)$$\sum_{i=1}^{\lceil \frac{n}{2}\rceil -1}d_{i}^{↓}<\sum_{i=n-\lceil \frac{n}{2}\rceil +1}^{n}d_{i}^{↓}$$
where $d_{i}^{↓}:={\left(d^{↓}\right)}_{i}$ is the *i*-th largest component of *d*, in the sense that *d* is well structured if and only if (Equation 5) holds. One way to observe this is to first reduce the well-structuredness of *d* to a family of inequalities $\sum_{i=1}^{\alpha }d_{i}^{↓}<\sum_{i=n-\alpha }^{n}d_{i}^{↓}$, α=1,…,n−1 (where α plays the role of |I| from Definition 1), and, in a second step, to realize that the inequality corresponding to $\alpha =\lceil \frac{n}{2}\rceil -1$ implies all other ones. Nonetheless, from (Equation 5), one sees that well-structuredness is a high-temperature phenomenon. Given the energies of the system $E_{1}\leq \ldots \leq E_{n}$ (without loss of generality, $E_{1}<E_{n}$ to avoid the trivial case of full degeneracy), there exists a unique critical temperature $T_{c}\geq 0$ such that $\sum_{i=1}^{\lceil \frac{n}{2}\rceil -1}e^{-E_{i}/T_{c}}=\sum_{i=n-\lceil \frac{n}{2}\rceil +1}^{n}e^{-E_{i}/T_{c}}$, and the corresponding Gibbs vector is well structured if and only if $T>T_{c}$. One way to prove this is to examine the auxiliary function $\phi :R_{+}\to R_{+}$ defined via
$$\phi \left(T\right):=\frac{\sum_{i=n-\lceil \frac{n}{2}\rceil +1}^{n}e^{-E_{i}/T}}{\sum_{i=1}^{\lceil \frac{n}{2}\rceil -1}e^{-E_{i}/T}}$$
and to see that $\lim_{T\to 0^{+}}\phi \left(T\right)\in \left[0,1\right]$, $\lim_{T\to \infty }\phi \left(T\right)>1$, and $\phi^{\prime }\left(T\right)>0$ for all T>0; the latter inequality follows at once from the readily verified expression
$$\phi^{\prime }\left(T\right)={\left(T\sum_{i=1}^{\lceil \frac{n}{2}\rceil -1}e^{-E_{i}/T}\right)}^{-2}\sum_{i=n-\lceil \frac{n}{2}\rceil +1}^{n}\sum_{j=1}^{\lceil \frac{n}{2}\rceil -1}\left(E_{i}-E_{j}\right)e^{-(E_{i}+E_{j})/T}$$
together with the observation that $E_{i}-E_{j}\geq 0$, because i>j, and even $E_{n}-E_{1}>0$ by assumption. Thus, by the intermediate value theorem, there exists a unique $T_{c}\geq 0$ such that $\phi (T_{c})=1$, and ϕ(T)>1 (which is equivalent to (Equation 5)) holds if and only if $T>T_{c}$.

Regardless, the notion of stable Gibbs states, as well as the fact that the well-structuredness of the Gibbs state appears if (and only if) the temperature exceeds a critical value, can be easily visualized via the standard simplex and the ordered Weyl chamber; cf. Figure 1.

Now, based on these notions, Loewy et al. were able to prove the following. Given any $d\in R_{++}^{n}$, the well-structuredness of *d* is equivalent to every extreme point of $s_{d}\left(n\right)$ (i.e., the Gibbs-stochastic matrices) being invertible (Theorem 3.1 [37]); the number of extreme points of $s_{d}\left(n\right)$ is maximal (when *d* is taken as a parameter) if and only if *d* is well structured and stable (Theorems 5.1 and 6.1 [37]); and, as a lower and upper bound to the number of extreme points of $s_{d}\left(n\right)$, they found $\left(n-1\right)!n^{n-2}$ and $n!n^{n-2}$, respectively. (Actually, they explained how to calculate an even better lower bound that grows asymptotically as $\left(n-1\right)!n^{n-2}\log n$ but it cannot be written down as well as the bound $\left(n-1\right)!n^{n-2}$.) The bound $\left(n-1\right)!n^{n-2}$ is a substantial improvement over the lower bound n! as first proven by Perfect and Mirsky [38].

## 3. Results

As we will see, the notions of well-structured and stable Gibbs states are key to answering fundamental questions in quantum thermodynamics. Not only do this definition and the associated, already known results (e.g., on extreme points of the Gibbs-stochastic matrices) carry over to our setting; this language is even suitable to solve seemingly unrelated problems in quantum thermodynamics and, subsequently, allows us to uncover new connections. Consequently, this section will feature two types of results: first, ones that are, at most, superficially concerned with the geometry of the thermomajorization polytope (Section 3.1), followed by some results on the geometric quantities (e.g., extreme points) of said polytope; cf. Section 3.2. The former can be seen as the general principles underlying quantum thermodynamics, while the latter are more state-dependent and more explicit in nature.

### 3.1. Cyclic State Transfers and Subspaces in Equilibrium

An overarching framework for this section is given by the notion of catalysis. There, in the most constrained form, one calls a state transition ρ↦ω
*strictly catalytic* if there exists an ancilla in state $\omega_{C}$ as well as an “allowed” operation Φ on the new overall system such that $\Phi \left(\rho ⊗\omega_{C}\right)=\omega ⊗\omega_{C}$ (although there are also “weaker” versions of catalytic transformations; cf. the review article [39]). For thermal operations in the quasi-classical realm, strict catalysis reduces to state transfers $x⊗z{≺}_{d}y⊗z$.

The idea behind such processes is of course that the catalyst $\omega_{C}$, resp. *z*, undergoes a cyclic process in order to be returned (uncorrelated and) unchanged. This raises fundamental questions such as, e.g., which cyclic processes are even possible in our thermodynamic framework, which properties such processes have, etc. This is what our first result concerns: in physical terms, it states that *global* cyclic thermodynamic processes are impossible for almost all temperatures (at least without access to external resources). While Theorem 1 below is concerned with two-step processes, as an immediate corollary, one obtains an analogous result for time-continuous cyclic processes. The precise statement here is that the impossibility of non-trivial cyclic processes is in one-to-one correspondence to the notion of stable Gibbs vectors.

**Theorem** **1.**
*Given $d\in R_{++}^{n}$, the following statements are equivalent:*


*(i)* 
*d is stable.*
*(ii)* 
*Given any $x,y\in R^{n}$, if $x{≺}_{d}y{≺}_{d}x$, then x=y.*


**Proof.** “(ii) ⇒ (i)”: We will prove this direction by contraposition, i.e., given *d* not stable, we construct x,y with x≠y such that $x{≺}_{d}y{≺}_{d}x$. Indeed, assume, to the contrary, that there exist I,J⊆{1,…,n}, I≠J such that $\sum_{i\in I}d_{i}=\sum_{j\in J}d_{j}$. Define $x:=\sum_{i\in I}d_{i}e_{i}$, $y:=\sum_{j\in J}d_{j}e_{j}$ and note that I≠J implies x≠y. We claim that $x{≺}_{d}y{≺}_{d}x$. Recalling Section 2.1, we prove this, equivalently, by showing that ${∥x-td∥}_{1}={∥y-td∥}_{1}$ for all t∈R:
$$\begin{array}{cc} {∥x-td∥}_{1} & =\sum_{i\in I}|d_{i}-td_{i}|\;+\sum_{i\in \{1,\ldots ,n\}\setminus I}\left|0-td_{i}\right|\\ \; & =\left(\sum_{i\in I}d_{i}\right)|1-t|\;+\left(e^{⊤}d-\sum_{i\in I}d_{i}\right)\left|t\right|\\ \; & =\left(\sum_{j\in J}d_{j}\right)|1-t|\;+\left(e^{⊤}d-\sum_{j\in J}d_{j}\right)\left|t\right|=\ldots ={∥y-td∥}_{1} \end{array}$$

“(i) ⇒ (ii)”: The idea of this part of the proof is that any vectors $x{≺}_{d}y{≺}_{d}x$ induce the same thermomajorization curve, and—because *d* is stable—applying this to the points where the curves change slope allows us to conclude x=y. More precisely, assume that *d* is stable and let $x,y\in R^{n}$ be given such that $x{≺}_{d}y{≺}_{d}x$. This implies $e^{⊤}x=e^{⊤}y$ and, more importantly, ${\mathrm{th}}_{d,x}\left(c\right)={\mathrm{th}}_{d,y}\left(c\right)$ for all $c\in [0,e^{⊤}d]$. However, these are piecewise linear functions characterized by elbow points, so, in particular, the points where ${\mathrm{th}}_{d,x},{\mathrm{th}}_{d,y}$ have a (non-trivial) change in slope coincide (recall Section 2.1). To be more precise, there exist k∈{1,…,n} and sets $I_{1}^{x},\ldots ,I_{k-1}^{x},I_{1}^{y},\ldots ,I_{k-1}^{y}\in P\left(\left\{1,\ldots ,n\right\}\right)$ such that

$\emptyset \neq I_{1}^{x}⊊I_{2}^{x}⊊\ldots ⊊I_{k-1}^{x}⊊\left\{1,\ldots ,n\right\}$ and $\emptyset \neq I_{1}^{y}⊊I_{2}^{y}⊊\ldots ⊊I_{k-1}^{y}⊊\left\{1,\ldots ,n\right\}$.${\mathrm{th}}_{d,x}$ changes slope precisely at the inputs $\left\{\sum_{i\in I_{l}^{x}}d_{i}\;:\;l=1,\ldots ,k-1\right\}$, and ${\mathrm{th}}_{d,y}$ changes slope precisely at $\left\{\sum_{i\in I_{l}^{y}}d_{i}\;:\;l=1,\ldots ,k-1\right\}$.${\mathrm{th}}_{d,x}\left(\sum_{i\in I_{l}^{x}}d_{i}\right)=\sum_{i\in I_{l}^{x}}x_{i}$ and ${\mathrm{th}}_{d,y}\left(\sum_{i\in I_{l}^{y}}d_{i}\right)=\sum_{i\in I_{l}^{y}}y_{i}$ for all l=1,…,k−1.

However, because the changes in slope coincide, this (due to d>0) shows $\sum_{i\in I_{l}^{x}}d_{i}=\sum_{i\in I_{l}^{y}}d_{i}$. By assumption, *d* is stable, so we obtain $I_{l}^{x}=I_{l}^{y}$ for all l=1,…,k−1. In particular,
$$\sum_{i\in I_{l}^{x}}x_{i}={\mathrm{th}}_{d,x}\left(\sum_{i\in I_{l}^{x}}d_{i}\right)={\mathrm{th}}_{d,y}\left(\sum_{i\in I_{l}^{x}}d_{i}\right)={\mathrm{th}}_{d,y}\left(\sum_{i\in I_{l}^{y}}d_{i}\right)=\sum_{i\in I_{l}^{y}}y_{i}=\sum_{i\in I_{l}^{x}}y_{i}$$
for all l=0,…,k when defining $I_{0}^{x}:=\emptyset =:I_{0}^{y}$ and $I_{k}^{x}:=\left\{1,\ldots ,n\right\}=:I_{k}^{y}$. Now that we have taken care of the points where the curves change slope, all that remain are the points in between. Consider any l=1,…,k. Because ${\mathrm{th}}_{d,y}$ is affine linear on $\left[\sum_{i\in I_{l-1}^{x}}d_{i},\sum_{i\in I_{l}^{x}}d_{i}\right]$ and the slope of ${\mathrm{th}}_{d,y}$ at “the” increment $d_{i}$ is given by $\frac{y_{i}}{d_{i}}$, there exists $c_{y,l}\in R$ such that $y_{i}=c_{y,l}\cdot d_{i}$ for all $i\in I_{l}^{x}\setminus I_{l-1}^{x}$; one argues analogously for ${\mathrm{th}}_{d,x}$ and obtains a constant $c_{x.l}$. Note that we use the stability of *d* here: the length $\sum_{i\in I_{l}^{x}\setminus I_{l-1}^{x}}d_{i}$ of the interval $\left[\sum_{i\in I_{l-1}^{x}}d_{i},\sum_{i\in I_{l}^{x}}d_{i}\right]$ can only come from adding up $\left\{d_{i}:i\in I_{l}^{x}\setminus I_{l-1}^{x}\right\}$. Now, because ${\mathrm{th}}_{d,x}$ and ${\mathrm{th}}_{d,y}$ coincide, by assumption, we find that $c_{x,l}=c_{y,l}\left(=:c_{l}\right)$ and thus
$$x=\sum_{i=1}^{n}x_{i}e_{i}=\sum_{l=1}^{k}\sum_{i\in I_{l}^{x}\setminus I_{l-1}^{x}}x_{i}e_{i}=\sum_{l=1}^{k}\sum_{i\in I_{l}^{x}\setminus I_{l-1}^{x}}\underset{=y_{i}}{\underset{︸}{c_{l}d_{i}}}e_{i}=y.$$

□

Of course, this result does not prohibit *local* cyclic processes, i.e., thermodynamic processes where only a subsystem returns to its original state at the end (so, precisely, catalysis). What Theorem 1 does assert, however, is that, in general, a quasi-classical cyclic process (modeled by thermal operations) that is *not* local has to use up some external resource along the way.

For the remainder of this section, our focus lies on (states on) subspaces that are “in equilibrium”. The motivation behind this notion is that if a state restricted to some subspace is a multiple of the Gibbs state, then all thermal operations on that subspace act trivially on it. Indeed, given a state *x* and a subspace *P* such that ${x|}_{P}$ is a multiple of the Gibbs vector, then any Gibbs-stochastic matrix that acts non-trivially only on *P*—i.e., it is of the form $A=A_{P}⊕1_{P^{⊥}}$—necessarily leaves *x* invariant. This of course generalizes to subsystems of coupled systems by choosing *P* appropriately. The precise definition reads as follows.

**Definition** **2.***Let $d\in R_{++}^{n}$, $y\in R^{n}$. We say that a subset P⊆{1,…,n}, |P|>1 of the system’s energy levels is* in equilibrium *if $\frac{y_{i}}{d_{i}}=\frac{y_{j}}{d_{j}}$ for all i,j∈P. On the other hand, if no such subset P satisfies this condition, we say that the system is in total non-equilibrium.*

In this language, our next result states that, regardless of whether there is a subspace in equilibrium (as long as the full system is not), every such subspace can be brought out of equilibrium by means of *d*-stochastic matrices. This, in particular, applies to catalytic state transfers: if, for example, a system is in equilibrium, then any catalyst (which itself is not in the Gibbs state) allows for bringing arbitrary energy levels of the original system out of equilibrium. The precise statement is derived via the dimension of the thermomajorization polytope and reads as follows.

**Theorem** **2.**
*Let $d\in R_{++}^{n}$, $y\in R^{n}$. The following statements are equivalent.*



*(i)* 
*$M_{d}\left(y\right)$ is not singular, i.e., $M_{d}\left(y\right)$ consists of more than merely y.*
*(ii)* 
*y is not a multiple of d.*
*(iii)* 
*The dimension of the convex polytope $M_{d}\left(y\right)$ is maximal, i.e., its dimension is n−1, which is equal to the dimension of the standard simplex $\Delta^{n-1}$ of all n-dimensional probability vectors.*

*In particular, if there exists a subset P⊊{1,…,n}, |P|>1 in equilibrium (i.e., $y_{i}/d_{i}=y_{j}/d_{j}$ for all i,j∈P), then there exists $z\in M_{d}\left(y\right)$ such that z is in total non-equilibrium.*


**Proof.** “(iii) ⇒ (i)”: Trivial. “(i) ⇒ (ii)”: Obvious via contrapositive. “(ii) ⇒ (iii)”: The dimension of $M_{d}\left(y\right)$ is trivially upper bounded by n−1 as it is a subset of the n−1-dimensional standard simplex (Corollary 17 [27]). For the converse, we argue by contraposition: if the dimension is strictly less than n−1, then there must exist a condition in (Equation 4) that is an equality (Chapter 8.2 [40]). More precisely, there must exist $m\in {\left\{0,1\right\}}^{n}$, $0<e^{⊤}m<n$ and c∈R such that $m^{⊤}x=c$ for all $x\in M_{d}\left(y\right)$. First, we determine *c*. Let $\sigma \in S_{n}$ be any permutation such that $\underline{\sigma }m$ is sorted non-increasingly, i.e., $\underline{\sigma }m={\left(1,\ldots ,1,0,\ldots ,0\right)}^{⊤}$. We know that $E_{d,y}\left(\sigma \right)\in M_{d}\left(y\right)$ so
$$\begin{array}{c} c=m^{⊤}E_{d,y}\left(\sigma \right)=\sum_{j=1}^{e^{⊤}m}{\left(E_{d,y}\left(\sigma \right)\right)}_{\sigma (j)}={\mathrm{th}}_{d,y}\left(\sum_{j=1}^{e^{⊤}m}d_{\sigma (j)}\right)={\mathrm{th}}_{d,y}\left(m^{⊤}d\right)\;. \end{array}$$
The final step is to evaluate the condition $m^{⊤}x={\mathrm{th}}_{d,y}\left(m^{⊤}d\right)$ at a multiple of *d*: because $\frac{de^{⊤}}{e^{⊤}d}\in s_{d}\left(n\right)$, one has $\frac{de^{⊤}}{e^{⊤}d}y=\frac{e^{⊤}y}{e^{⊤}d}d\in M_{d}\left(y\right)$. Therefore,
(6)$$\begin{array}{cc} {\mathrm{th}}_{d,y}\left(m^{⊤}d\right)=c=m^{⊤}\left(\frac{e^{⊤}y}{e^{⊤}d}d\right) & =\frac{e^{⊤}d-m^{⊤}d}{e^{⊤}d}\cdot 0+\frac{m^{⊤}d}{e^{⊤}d}e^{⊤}y\\ \; & =\frac{e^{⊤}d-m^{⊤}d}{e^{⊤}d-0}{\mathrm{th}}_{d,y}\left(0\right)+\frac{m^{⊤}d-0}{e^{⊤}d-0}{\mathrm{th}}_{d,y}\left(e^{⊤}d\right)\;. \end{array}$$
Because ${\mathrm{th}}_{d,y}$ is concave and because $m^{⊤}d\in \left(0,e^{⊤}d\right)$ by the assumptions on *m* and *d*, Lemma A3 (iv) (Appendix B) shows that (Equation 6) can only hold if ${\mathrm{th}}_{d,y}$ is linear. However, the latter is equivalent to *y* being a multiple of *d* as, by definition, the slopes of ${\mathrm{th}}_{d,y}$ are given by $y_{j}/d_{j}$.Finally, the additional statement follows at once from the following two facts: (a) the collection of all vectors in total non-equilibrium is dense in $\Delta^{n-1}$, and (b) $M_{d}\left(y\right)$ contains an interior point with regard to the hyperplane $W_{e}:=\left\{z\in R^{n}:e^{⊤}z=1\right\}$. While (b) is due to (iii), for (a), note that, given any P⊊{1,…,n}, |P|>1 the set of vectors $y\in W_{e}$ in equilibrium (with regard to *P*) forms a lower-dimensional subspace of $W_{e}$. In particular, this set is nowhere dense (i.e., its closure has an empty interior [41]), which continues to hold when taking the union over all (finitely many) such *P*. However, this, in particular, implies that the complement of this set—i.e., the collection of all vectors in total non-equilibrium—is dense in $W_{e}$. This concludes the proof. □

Returning to the language of subspaces in equilibrium, note that this result is an existence result and does not indicate anything about the potential “amplitude” of such transfers. Mathematically, Theorem 2 complements the old result of Hartfiel that the dimension of $s_{d}\left(n\right)$ for all d>0 is ${\left(n-1\right)}^{2}$ (Theorem 3.1 [30]).

**Remark** **3.**
*The assumption d>0 in Theorem 2 is necessary as hinted at by the fact that $\dim(s_{d}\left(n\right))$ for a general $d\in R_{+}^{n}$ depends on the number of zeros in d (Theorem 3.2 [30]). This is well illustrated by the simple example $d={\left(1,1,0\right)}^{⊤}$, $y={\left(1,0,0\right)}^{⊤}$ because then $M_{d}\left(y\right)=\left\{{\left(c,1-c,0\right)}^{⊤}\;:\;c\in \left[0,1\right]\right\}$, which is not two- but only one-dimensional.*


### 3.2. Extreme Points of the Thermomajorization Polytope

Let us stress that—until now—the concept of subspaces in equilibrium (and hence Theorem 2) has been logically independent from the notions of stability and well-structuredness. This missing connection will be established below, where well-structured states and subspaces in equilibrium will be linked via the geometric properties of the thermomajorization polytope—in particular, the number of its extreme points. (An analytic form of the extreme points of $M_{d}\left(y\right)$ has appeared independently in the physics [8,15] and the mathematics [27] literature.) The key mathematical object for doing so is the extreme point map $E_{d,y}$, which is defined as follows.

**Definition** **3.***Let $d\in R_{++}^{n}$, $y\in R^{n}$. Define the* extreme point map *$E_{d,y}$ on the symmetric group $S_{n}$ via*
$$E_{d,y}:S_{n}\to R^{n},\;\sigma \mapsto {\left({\mathrm{th}}_{d,y}\left(\sum_{i=1}^{\sigma^{-1}\left(j\right)}d_{\sigma (i)}\right)-{\mathrm{th}}_{d,y}\left(\sum_{i=1}^{\sigma^{-1}\left(j\right)-1}d_{\sigma (i)}\right)\right)}_{j=1}^{n}\;.$$
*Equivalently, ${\left(\underline{\sigma }E_{d,y}\left(\sigma \right)\right)}_{j}={\left(E_{d,y}\left(\sigma \right)\right)}_{\sigma (j)}={\mathrm{th}}_{d,y}\left(\sum_{i=1}^{j}d_{\sigma (i)}\right)-{\mathrm{th}}_{d,y}\left(\sum_{i=1}^{j-1}d_{\sigma (i)}\right)$, where $\underline{\sigma }$ is the permutation matrix corresponding to σ as defined above (Remark 1).*

As the name suggests, for all $d\in R_{++}^{n}$ and $y\in R^{n}$, the image of $E_{d,y}$ equals the set $\mathrm{ext}(M_{d}\left(y\right))$ of extreme points of $M_{d}\left(y\right)$, and thus $M_{d}\left(y\right)=\mathrm{conv}\left\{E_{d,y}\left(\sigma \right):\sigma \in S_{n}\right\}$ (Theorem 16 [27]). It should be noted that the extreme point property also manifests in the thermomajorization curves, relating to the concept of *tight thermomajorization*. Given any $y,z\in R_{+}^{n}$, the point *z* is an extreme point of $M_{d}\left(y\right)$ if and only if all elbow points of ${\mathrm{th}}_{d,z}$ lie on the curve ${\mathrm{th}}_{d,y}$ (Theorem 2 [17]). Moreover, there is a strong connection between the extreme points of $M_{d}\left(y\right)$ and a special class of extreme points of the Gibbs-stochastic matrices, which we will elaborate on at the end of this section. Moreover, Section 4.1 below presents a step-by-step calculation of the extreme point map and explains how it, equivalently, can be computed graphically using the thermomajorization curve.

Our focus for now, however, lies on the properties of the map $E_{d,y}$ and geometric aspects of $M_{d}\left(y\right)$. From Definition 3, it is clear that the maximal number of extreme points of $M_{d}\left(y\right)$ is $n!=|S_{n}|$; if there are strictly fewer than n! extreme points, we say that the polytope is *degenerate*. The goal of this section is to prove the following result, which states that the degeneracy of the polytope is a “witness” for subspaces in equilibrium, assuming that *d* is well structured (equivalently, assuming large enough temperatures; cf. Section 2.2).

**Theorem** **3.**
*Let $d\in R_{++}^{n}$, $y\in R^{n}$. If $E_{d,y}$ is not injective, i.e., $|\mathrm{ext}(M_{d}\left(y\right))|<n!$, then at least one of the following holds:*


*(i)* 
*y has a subspace which is in equilibrium with respect to d.*
*(ii)* 
*d is not well structured, i.e., the temperature is below the critical value $T\leq T_{c}$; cf. Section 2.2.*


Note that in the example of Section 4.1, degenerate extreme points occur, and the degeneracy stems from the fact that there exists a subspace in equilibrium. This can be shown generally as specifying the preimage of *y* under the extreme point map $E_{d,y}$ is found to be straightforward.

**Lemma** **1.**
*Let $d\in R_{++}^{n}$, $y\in R^{n}$, and $\sigma \in S_{n}$ be given. One has $E_{d,y}\left(\sigma \right)=y$ if and only if $\frac{y_{\sigma (1)}}{d_{\sigma (1)}}\geq \ldots \geq \frac{y_{\sigma (n)}}{d_{\sigma (n)}}$.*


**Proof.** Assume that $\sigma \in S_{n}$ satisfies $E_{d,y}\left(\sigma \right)=y$ so ${\left(E_{d,y}\left(\sigma \right)\right)}_{\sigma (j)}=y_{\sigma (j)}$ for all j=1,…,n. By definition of $E_{d,y}$, this means ${\mathrm{th}}_{d,y}\left(\sum_{i=1}^{j}d_{\sigma (i)}\right)={\mathrm{th}}_{d,y}\left(\sum_{i=1}^{j-1}d_{\sigma (i)}\right)+y_{\sigma (j)}$, which, by induction, is equivalent to ${\mathrm{th}}_{d,y}\left(\sum_{i=1}^{j}d_{\sigma (i)}\right)=\sum_{i=1}^{j}y_{\sigma (i)}$ for all j=1,…,n−1. However, by Lemma A2 (Appendix B), this holds if and only if $y_{\sigma (1)}/d_{\sigma (1)}\geq \ldots \geq y_{\sigma (n)}/d_{\sigma (n)}$. □

Clearly, then, the extreme point *y* is degenerate (in the sense that it has multiple preimages under $E_{d,y}$) if and only if there exists a subspace that is in equilibrium; cf. Definition 2. The converse, however, is not true. Indeed, the example in Section 4.2 shows that degenerate extreme points can occur even if the system is in total non-equilibrium. Note, however, that, in this example, the vector *d* is not well structured, indicating a low temperature, as required by Theorem 3.

Note that Lemma 1 relates to the concept of virtual temperatures [6,42,43,44] as multiple permutations are mapped to *y* under $E_{d,y}$ if and only if there exist i,j∈{1,…,n} such that the background temperature equals $T=\frac{E_{j}-E_{i}}{\ln\left(y_{i}\right)-\ln\left(y_{j}\right)}=:T_{ij}$ (which is equivalent to $\frac{y_{i}}{d_{i}}=\frac{y_{j}}{d_{j}}$). In other words, virtual temperatures characterize when another corner of the polytope “crosses” the initial state. This also relates to the notion of passivity: the degeneracy of $M_{d}\left(y\right)$ at temperature $T_{ij}$ corresponding to the transition between $E_{i}$ and $E_{j}$ is physical (i.e., $T_{ij}>0$) only if the initial state diag(y) is passive, meaning that no work can be extracted via unitary transformations (Section III [43]).

**Example** **1.**
*It is worth addressing the case of classical majorization, i.e., d=e (up to a factor that is of no consequence). We find that ${\mathrm{th}}_{d,y}\left(\sum_{i=1}^{j}d_{\sigma (i)}\right)=\sum_{i=1}^{j}y_{i}^{↓}$ for any j=1,…,n, $\sigma \in S_{n}$; this recovers $E_{d,y}\left(\sigma \right)={\underline{\sigma }}^{-1}y^{↓}$ (Chapter 4, Proposition C.1 [28]). Therefore, degeneracies of $M_{e}\left(y\right)$ correspond to some entries of the initial state coinciding: given any $y\in R^{n}$, the number of extreme points of $M_{e}\left(y\right)$ equals $n!/(p_{1}!\cdot \ldots \cdot p_{m}!)$, where ${\left(p_{i}\right)}_{i=1}^{m}$ denote the multiplicities of y, i.e., $y^{↓}=\sum_{i=1}^{m}y_{i}^{↓}\sum_{j=1}^{p_{i}}e_{p_{1}+\ldots +p_{i-1}+j}$ where $y_{1}^{↓}>\ldots >y_{m}^{↓}$.*
*This, in fact, implies the “uniformity” of the classical majorization polytope’s degeneracy in the sense that the preimage of* each *extreme point under $E_{d,y}$ has the same size, which is certainly false for general $d\in R_{++}^{n}$; cf. Section 4.1 and, in particular, Table 1. This uniformity is related to the fact that $s_{e}\left(n\right)$ contains all permutation matrices, which yields a group action of $S_{n}$ on $s_{e}\left(n\right)$. This group action is transitive on the vertices, and hence all vertices are equivalent. This does not hold for general $d\in R_{++}^{n}$: the inverse of some invertible element of $s_{d}\left(n\right)$ is again in $s_{d}\left(n\right)$ if and only if it is a permutation matrix (Remark 4.1 [45]).*

Reviewing the construction of extreme points in Section 4.1, and especially the graphical approach, one notices quickly how degeneracies can occur; cf. in particular Remark 5. Indeed, given $d\in R_{++}^{n}$, $y\in R^{n}$, there exists m∈{0,…,n−1} as well as real numbers $\Delta_{0},\ldots ,\Delta_{m+1}$ such that $0=\Delta_{0}<\Delta_{1}<\ldots <\Delta_{m}<\Delta_{m+1}=e^{⊤}d$ and ${\mathrm{th}}_{d,y}$ changes slope at an input $x\in (0,e^{⊤}d)$ if and only if $x=\Delta_{k}$ for some k=1,…,m. The example in Section 4.1 shows that degeneracies occur when two or more intervals of length $d_{i}$, i=1,…,m (ordered according to σ) are contained in the same interval $\left(\Delta_{k-1},\Delta_{k}\right)$.

This motivates the following definition. For $\sigma \in S_{n}$, k∈{1,…,m+1}, define
(7)$$I_{k}^{\sigma }:=\left\{j\in \left\{1,\ldots ,n\right\}\;:\;\left[\sum_{i=1}^{\sigma^{-1}\left(j\right)-1}d_{\sigma (i)},\sum_{i=1}^{\sigma^{-1}\left(j\right)}d_{\sigma (i)}\right]\cap \left(\Delta_{k-1},\Delta_{k}\right)\neq \emptyset \right\}\;.$$
The elements of $I_{k}^{\sigma }$ are those j∈{1,…,n} for which the interval of length $d_{j}$ corresponding to the partition induced by σ intersects $\left(\Delta_{k-1},\Delta_{k}\right)$. In particular, ${⋃}_{k=1}^{m+1}I_{k}^{\sigma }=\left\{1,\ldots ,n\right\}$ for all $\sigma \in S_{n}$. An illustrative example is given in Section 4.4.

The importance of this definition comes from its ability to characterize the image of the extreme point map, as the following result shows.

**Proposition** **1.**
*Let $d\in R_{++}^{n}$, $y\in R^{n}$ and let m∈{1,…,n−1} be the number of changes in the slope of ${\mathrm{th}}_{d,y}$. Given any $\sigma ,\tau \in S_{n}$, one has $E_{d,y}\left(\sigma \right)=E_{d,y}\left(\tau \right)$ if and only if $I_{k}^{\sigma }=I_{k}^{\tau }$ for all k=1,…,m+1.*


**Proof.** “⇒”: Assume $E_{d,y}\left(\sigma \right)=E_{d,y}\left(\tau \right)$. Then,
(8)$${\mathrm{th}}_{d,y}\left(\sum_{i=1}^{\sigma^{-1}\left(j\right)}d_{\sigma (i)}\right)-{\mathrm{th}}_{d,y}\left(\sum_{i=1}^{\sigma^{-1}\left(j\right)-1}d_{\sigma (i)}\right)={\mathrm{th}}_{d,y}\left(\sum_{i=1}^{\tau^{-1}\left(j\right)}d_{\tau (i)}\right)-{\mathrm{th}}_{d,y}\left(\sum_{i=1}^{\tau^{-1}\left(j\right)-1}d_{\tau (i)}\right)$$
for all j=1,…,n; see Definition 3. Now, given j∈{1,…,n}, there are two (non-exclusive) possibilities: either $\left[\sum_{i=1}^{\sigma^{-1}\left(j\right)-1}d_{\sigma (i)},\sum_{i=1}^{\sigma^{-1}\left(j\right)}d_{\sigma (i)}\right]=\left[\sum_{i=1}^{\tau^{-1}\left(j\right)-1}d_{\tau (i)},\sum_{i=1}^{\tau^{-1}\left(j\right)}d_{\tau (i)}\right]$, which implies $j\in I_{k}^{\sigma }$ if and only if $j\in I_{k}^{\tau }$, or the two intervals do not coincide. The latter implies that ${\mathrm{th}}_{d,y}$ is affine linear on $\mathrm{conv}(\left[\sum_{i=1}^{\sigma^{-1}\left(j\right)-1}d_{\sigma (i)},\sum_{i=1}^{\sigma^{-1}\left(j\right)}d_{\sigma (i)}\right]\cup \left[\sum_{i=1}^{\tau^{-1}\left(j\right)-1}d_{\tau (i)},\sum_{i=1}^{\tau^{-1}\left(j\right)}d_{\tau (i)}\right])$ by our argument from above. However, this means that both these intervals have to be contained in the same interval $\left[\Delta_{k-1},\Delta_{k}\right]$; hence, $j\in I_{k}^{\sigma }$ and $j\in I_{k}^{\tau }$.“⇐”: Assume by contraposition that $E_{d,y}\left(\sigma \right)\neq E_{d,y}\left(\tau \right)$ so there exists j=1,…,n such that (Equation 8) does not hold. Thus, $J_{\sigma }:=\left[\sum_{i=1}^{\sigma^{-1}\left(j\right)-1}d_{\sigma (i)},\sum_{i=1}^{\sigma^{-1}\left(j\right)}d_{\sigma (i)}\right]$ has to differ from $J_{\tau }:=\left[\sum_{i=1}^{\tau^{-1}\left(j\right)-1}d_{\tau (i)},\sum_{i=1}^{\tau^{-1}\left(j\right)}d_{\tau (i)}\right]$ (which is only possible if $\sum_{i=1}^{\sigma^{-1}\left(j\right)-1}d_{\sigma (i)}\neq \sum_{i=1}^{\tau^{-1}\left(j\right)-1}d_{\tau (i)}$ because the intervals have the same length). Moreover, ${\mathrm{th}}_{d,y}$ cannot be affine linear on $\mathrm{conv}(J_{\sigma }\cup J_{\tau })$ so there exists k∈{1,…,m} such that the change in slope $\Delta_{k}$ lies in the interior of $\mathrm{conv}(J_{\sigma }\cup J_{\tau })$.Now, there are two possible cases (cf. Figure 2): if $\Delta_{k}\notin J_{\sigma }\cap J_{\tau }$, then $\Delta_{k}\notin J_{\sigma }$ (or $\Delta_{k}\notin J_{\tau }$) so $\mathrm{int}(J_{\sigma })$ ($\mathrm{int}(J_{\tau })$) is to the left or to the right of $\Delta_{k}$. Either way, there exists $\tilde{k}\in \left\{1,\ldots ,m+1\right\}$ such that $j\in I_{\tilde{k}}^{\tau }$ but $j\notin I_{\tilde{k}}^{\sigma }$ so we are finished. For the second case, assume that $\Delta_{k}\in J_{\sigma }\cap J_{\tau }$ and, without loss of generality, that $\min J_{\sigma }<\min J_{\tau }$, i.e., $\sum_{i=1}^{\sigma^{-1}\left(j\right)-1}d_{\sigma (i)}<\sum_{i=1}^{\tau^{-1}\left(j\right)-1}d_{\tau (i)}$. Therefore, one can find $\alpha \in \{1,\ldots ,\tau^{-1}\left(j\right)-1\}$ such that $\tau \left(\alpha \right)\notin \{\sigma \left(1\right),\ldots ,\sigma \left(\sigma^{-1}\left(j\right)-1\right),j\}$. This in turn implies $\sigma^{-1}\left(\tau \left(\alpha \right)\right)>\sigma^{-1}\left(j\right)$, i.e., the interval corresponding to $d_{\tau (\alpha )}$ is to the left of $J_{\tau }⊇J_{\tau }\cap J_{\sigma }$ but to the right of $J_{\sigma }⊇J_{\sigma }\cap J_{\tau }$. Because $\Delta_{k}\in J_{\sigma }\cap J_{\tau }$, this yields $\tilde{k}\in \left\{1,\ldots ,k\right\}$ such that $\tau \left(\alpha \right)\in I_{\tilde{k}}^{\tau }$ but $\tau \left(\alpha \right)=\sigma \left(\sigma^{-1}\left(\tau \left(\alpha \right)\right)\right)\notin I_{\tilde{k}}^{\sigma }$. This concludes the proof. □

This means that σ and τ yield the same extreme point if and only if they differ exactly by a permutation of the intervals of length $d_{i}$ such that each interval remains in the same region where ${\mathrm{th}}_{d,y}$ is affine linear. Note that the case of classical majorization follows from this since, in that case, such permutations differ exactly by the elements of the stabilizer of *y*. Similarly, the result about preimages of *y* (Lemma 1) follows from this.

Nonetheless, Proposition 1 gives a simple criterion to check whether the images of different permutations under the extreme point map $E_{d,y}$ coincide or not. In particular, a bound on the degeneracy of any extreme point can be given by how many of the $d_{i}$ intervals fit into the same $\left[\Delta_{k-1},\Delta_{k}\right]$ interval. This is related to the bin packing problem in computer science, which (is strongly NP-complete but) admits some reasonable approximations. Another way to look at the above results is that $|\mathrm{ext}\left(M_{d}\left(y\right)\right)\left|\leq n!+1-\right|{\left(S_{n}\right)}_{y/d}|$ by Lemma 1, where ${\left(S_{n}\right)}_{y/d}$ is the stabilizer of the vector $y/d={\left(y_{i}/d_{i}\right)}_{i=1}^{n}$ in $S_{n}$, and the examples in Section 4.1 and Section 4.2 show that this bound is not tight. These examples suggest that improving this bound via an analytic expression is a non-trivial task. Finally, Proposition 1 allows us to prove Theorem 3.

**Proof** **of** **Theorem 3.**Assume that there exist distinct $\sigma ,\tau \in S_{n}$ such that $E_{d,y}\left(\sigma \right)=E_{d,y}\left(\tau \right)$. Then, by Proposition 1, σ and τ differ by a permutation such that each *d*-interval remains in the same affine linear region of the thermomajorization curve. If $i\mapsto \frac{y_{i}}{d_{i}}$ is injective, this means that there have to exist pairwise distinct i,j,k∈{1,…,n} such that both the intervals corresponding to $d_{j},d_{k}$ “fit inside” $d_{i}$. Therefore, $d_{i}\geq d_{j}+d_{k}$, meaning that *d* cannot be well structured. □

Note that the proof of Theorem 3 actually shows that $d_{i}\geq d_{j}+d_{k}$ for some distinct i,j,k, which is a property stronger than the lack of well-structuredness. We wish to stress that while condition (i) of Theorem 3 ensures the degeneracy of $M_{d}\left(y\right)$ (Lemma 1), the lack of well-structuredness of *d* is not sufficient for $M_{d}\left(y\right)$ to be degenerate. This phenomenon is easy to understand in the graphical representation: even if it holds that $d_{i}\geq d_{j}+d_{k}$ for some distinct i,j,k—and hence *d* is not well-structured—it might happen that there is no permutation of the intervals that achieves the degeneracy. An example of this is given in Section 4.3.

Let us conclude this section by examining the operator lift. More precisely, due to $M_{d}\left(y\right)=\mathrm{conv}\left\{Ay:A\in \mathrm{ext}\left(s_{d}\left(n\right)\right)\right\}$ (Chapter 14, Section C, C.2. Observations (iii) [28]), Minkowski’s theorem (Theorem 5.10 [46]) shows that, given any extreme point *z* of $M_{d}\left(y\right)$, there exists an extreme point *A* of $s_{d}\left(n\right)$ such that z=Ay. Now, the obvious question is whether, given some extreme point of $M_{d}\left(y\right)$, there is an easy way to recover one (or every) process that maps the initial state to the point in question. While, given any initial and any final state, there already exists an algorithm to construct a Gibbs-stochastic matrix mapping the former to the latter [22], it emerges that if the final state is an extreme point, then this procedure simplifies considerably.

**Definition** **4.**
*Given $d\in R_{++}^{n}$ and permutations $\sigma ,\tau \in S_{n}$, there exists, for all j=1,…,n−1, a unique $\alpha_{j}\in \left\{1,\ldots ,n\right\}$ such that $\sum_{i=1}^{j}d_{\sigma (i)}\in \left(\sum_{i=1}^{\alpha_{j}-1}d_{\tau (i)},\sum_{i=1}^{\alpha_{j}}d_{\tau (i)}\right]$. Moreover, set $\alpha_{0}:=1$ and $\alpha_{n}:=n$. Based on this, define a matrix $A_{\sigma \tau }\in R_{+}^{n\times n}$ via*

(9)
$${\left(A_{\sigma \tau }\right)}_{\sigma (j)\tau (k)}:=\left\{\begin{array}{cc} \left(\sum_{i=1}^{\alpha_{j-1}}d_{\tau (i)}-\sum_{i=1}^{j-1}d_{\sigma (i)}\right)\cdot d_{\tau (\alpha_{j-1})}^{-1} & if\;k=\alpha_{j-1}<\alpha_{j}\\ 1 & if\;\alpha_{j-1}<k<\alpha_{j}\\ \left(\sum_{i=1}^{j}d_{\sigma (i)}-\sum_{i=1}^{\alpha_{j}-1}d_{\tau (i)}\right)\cdot d_{\tau (\alpha_{j})}^{-1} & if\;\alpha_{j-1}<\alpha_{j}=k\\ \frac{d_{\sigma (j)}}{d_{\tau (\alpha_{j})}} & if\;\alpha_{j-1}=\alpha_{j}=k\\ 0 & else \end{array}\right.$$

*for all j,k=1,…,n.*


This object has already appeared in the literature as “β-permutation” [8] and it coincides with the concept of a “biplanar extremal transportation matrix” [17] (up to the isomorphism X↦Xdiag(d) from Section 2.2). The latter name, rightly, suggests that the matrix $A_{\sigma \tau }$ for all $y\in R^{n}$, $d\in R_{++}^{n}$ and all $\sigma ,\tau \in S_{n}$ is an extreme point of the Gibbs-stochastic matrices (Theorem 1 ff. [17]). However—due to the lower bound $\left(n-1\right)!n^{n-2}$ on the number of extreme points of $s_{d}\left(n\right)$ from Section 2.2—for n≥4, there must exist extreme points of $s_{d}\left(n\right)$ that are not of the form $A_{\sigma \tau }$ (i.e., they are not a β-permutation). Actually, the lower bound in question shows that, for large *n*, “almost no” extreme point of $s_{d}\left(n\right)$ is a β-permutation as ${\left(n!\right)}^{2}/\left(\left(n-1\right)!n^{n-2}\right)\to 0$ as n→∞ (Section IV.B [17]).

Moreover, and more importantly, if τ is chosen such that $\frac{y_{\tau (1)}}{d_{\tau (1)}}\geq \ldots \geq \frac{y_{\tau (n)}}{d_{\tau (n)}}$, then $A_{\sigma \tau }$ maps the initial state *y* to the extreme point $E_{d,y}\left(\sigma \right)$, and if y/d is non-degenerate, then $A_{\sigma \tau }$ is the *unique* Gibbs-stochastic matrix that maps *y* to $E_{d,y}\left(\sigma \right)$; cf. [8], (Lemma 3 [17]). Not only does this yield a simple way to reverse-engineer a process that generates an extreme point in question; it also constitutes an alternative means to evaluate the extreme point map $E_{d,y}$ from Definition 3.

**Remark** **4.**
*Given a permutation σ, a matrix $A_{\sigma \tau }$ (stored in a sparse matrix format) can be constructed algorithmically in, at most, O(nlogn) steps using Definition 4, as the limiting step is to find an appropriate permutation τ. Any algorithm that computes a process matrix A for an arbitrary state transfer, including the one given in [22], must have the worst time complexity of at least $\Omega (n^{2})$, since A is dense in general. Hence, the structure of the $A_{\sigma \tau }$ leads to an improved runtime for the special case where the final state is extremal.*


Now, our main contribution to this concept reads as follows. While Definition 4 (which matches the definition given by [8]) as well as Mazurek’s construction for biplanar extremal transportation matrices appear rather convoluted, we will present a very simple construction of this matrix in Section 4.5 below. The only required step is to compare the sets $\left\{\sum_{i=1}^{j}d_{\sigma (i)}:j=1,\ldots ,n\right\}$ and $\left\{\sum_{i=1}^{j}d_{\tau (i)}:j=1,\ldots ,n\right\}$, which, en passant, reaffirms the observation made by Alhambra et al. that Definition 4 is independent of the initial state *y*. Note that these ideas are closely related to the calculation of extreme points given in Section 4.1 and to the index sets $I_{k}^{\sigma }$ defined in (Equation 7), which are the main concepts in the proof of Proposition 1.

## 4. Detailed Examples

The objects introduced in Section 3.2 can be computed explicitly and they have simple graphical interpretations, e.g., via thermomajorization curves. The following examples show this in detail.

### 4.1. Extreme Point Map

Definition 3 contains a simple algorithm to evaluate the extreme point map $E_{d,y}\left(\sigma \right)$. Given some permutation $\sigma \in S_{n}$, find the value of the thermomajorization curve at $d_{\sigma (1)},d_{\sigma (1)}+d_{\sigma (2)},\ldots ,\sum_{i=1}^{n-1}d_{\sigma (i)}$, take the difference in consecutive values, and arrange them into a vector that is ordered according to σ. Let us provide a detailed example.

Let $y={\left(4,0,1\right)}^{⊤}$ and $d={\left(4,2,1\right)}^{⊤}$. One verifies ${\mathrm{th}}_{d,y}\left(c\right)=\min\left\{c,5\right\}$ for all c∈[0,7] by direct computation; cf. Figure 3 below. Now, let $\sigma \in S_{3}$ be the permutation σ(1)=2, σ(2)=3, and σ(3)=1; in two-line notation, this reads $\sigma =\left(\begin{array}{ccc} 1 & 2 & 3\\ 2 & 3 & 1 \end{array}\right)$ (henceforth, σ=(231) for short). Our goal is to compute the extreme point $E_{d,y}\left(\sigma \right)$ of $M_{d}\left(y\right)$ that “corresponds” to σ. We will use the second formulation provided in Definition 3. First,
$$\begin{array}{cc} {\left(E_{d,y}\left(\sigma \right)\right)}_{2}={\left(E_{d,y}\left(\sigma \right)\right)}_{\sigma (1)} & ={\mathrm{th}}_{d,y}\left(d_{\sigma (1)}\right)-{\mathrm{th}}_{d,y}\left(0\right)={\mathrm{th}}_{d,y}\left(d_{2}\right)=\min\left\{2,5\right\}=2\;, \end{array}$$
followed by ${\left(E_{d,y}\left(\sigma \right)\right)}_{3}={\left(E_{d,y}\left(\sigma \right)\right)}_{\sigma (2)}={\mathrm{th}}_{d,y}\left(d_{\sigma (1)}+d_{\sigma (2)}\right)-{\mathrm{th}}_{d,y}\left(d_{\sigma (1)}\right)=\min\left\{3,5\right\}-\min\left\{2,5\right\}=1$ and analogously for ${\left(E_{d,y}\left(\sigma \right)\right)}_{1}$. Thus, $E_{d,y}\left(\sigma \right)={\left(2,2,1\right)}^{⊤}$. This procedure can be easily visualized; cf. Figure 3.

The full image of $E_{d,y}$ is computed analogously: one finds (cf. Table 1)
$$\mathrm{ext}\left(M_{d}\left(y\right)\right)=\left\{\left(\begin{array}{c} 4\\ 1\\ 0 \end{array}\right),\left(\begin{array}{c} 4\\ 0\\ 1 \end{array}\right),\left(\begin{array}{c} 2\\ 2\\ 1 \end{array}\right),\left(\begin{array}{c} 3\\ 2\\ 0 \end{array}\right)\right\}\;.$$

With this in mind, let us reformulate the definition of the map $E_{d,y}\left(\sigma \right)$ to obtain an even better understanding.

**Remark** **5.**
*Given $d\in R_{++}^{n}$ and $\sigma \in S_{n}$, the permutation σ indicates how to order the segments of length $d_{i}$. These can be visualized as lying head to tail on the x-axis, i.e., as a tiling of the interval $\left[0,e^{⊤}d\right]$. The contact points between the intervals are then $d_{\sigma (1)}$, $d_{\sigma (1)}+d_{\sigma (2)}$, and so on. Now, we evaluate ${\mathrm{th}}_{d,y}$ at these points and look at the corresponding increments ${\left(E_{d,y}\left(\sigma \right)\right)}_{\sigma (j)}={\mathrm{th}}_{d,y}\left(\sum_{i=1}^{j}d_{\sigma (i)}\right)-{\mathrm{th}}_{d,y}\left(\sum_{i=1}^{j-1}d_{\sigma (i)}\right)$ for j=1,…,n, as visualized in Figure 3. This construction relates to the previously mentioned notion of tight thermomajorization, as these contact points, in turn, are the elbow points of the thermomajorization curve of $E_{d,y}\left(\sigma \right)$.*

*Note that, by Definition 3, ${\left(E_{d,y}\left(\sigma \right)\right)}_{\sigma (j)}$ is the increment in ${\mathrm{th}}_{d,y}$ over a distance of length $d_{\sigma (j)}$. In particular, for any j=1,…,n, the entry ${\left(E_{d,y}\left(\sigma \right)\right)}_{j}$ corresponds to the increment over the interval $d_{j}$, regardless of where it is in our tiling. Hence, two permutations σ,τ give the same extreme point $E_{d,y}\left(\sigma \right)=E_{d,y}\left(\tau \right)$ if and only if each interval $d_{j}$ yields the same increment in ${\mathrm{th}}_{d,y}$ for both permutations. In the example above, this happens because the vector y/d is degenerate. The following example shows, however, that this is not necessary.*


### 4.2. Degeneracy

Now, we present a different example: consider $d={\left(1,2,10\right)}^{⊤}$, $y={\left(1,4,5\right)}^{⊤}$. The key insight is that even though $\frac{y}{d}={\left(1,2,0.5\right)}^{⊤}$ is non-degenerate, the polytope $M_{d}\left(y\right)$ appears to be degenerate. The reason that this is allowed to happen is that *d* is not well structured. Indeed, as in Section 4.1, one computes
(10)$${\mathrm{th}}_{d,y}\left(c\right)=\left\{\begin{array}{cc} 2c & c\in [0,2]\\ c+2 & c\in [2,3]\\ 0.5c+3.5 & c\in [3,13] \end{array}\right.$$
for all c∈[0,13] and hence one finds
$$E_{d,y}\left(3\;1\;2\right)=\left(\begin{array}{c} {\mathrm{th}}_{d,y}\left(11\right)-{\mathrm{th}}_{d,y}\left(10\right)\\ {\mathrm{th}}_{d,y}\left(13\right)-{\mathrm{th}}_{d,y}\left(11\right)\\ {\mathrm{th}}_{d,y}\left(10\right) \end{array}\right)=\left(\begin{array}{c} 9-8.5\\ 10-9\\ 8.5 \end{array}\right)=\left(\begin{array}{c} 0.5\\ 1\\ 8.5 \end{array}\right)$$
and
$$E_{d,y}\left(3\;2\;1\right)=\left(\begin{array}{c} {\mathrm{th}}_{d,y}\left(13\right)-{\mathrm{th}}_{d,y}\left(12\right)\\ {\mathrm{th}}_{d,y}\left(12\right)-{\mathrm{th}}_{d,y}\left(10\right)\\ {\mathrm{th}}_{d,y}\left(10\right) \end{array}\right)=\left(\begin{array}{c} 10-9.5\\ 9.5-8.5\\ 8.5 \end{array}\right)=\left(\begin{array}{c} 0.5\\ 1\\ 8.5 \end{array}\right)\;.$$
Despite the two permutations differing, their image under $E_{d,y}$ coincides. This comes from the fact that ${\mathrm{th}}_{d,y}\left(10\right)+{\mathrm{th}}_{d,y}\left(13\right)={\mathrm{th}}_{d,y}\left(11\right)+{\mathrm{th}}_{d,y}\left(12\right)$ because ${\mathrm{th}}_{d,y}{|}_{\left[3,13\right]}$ is affine linear; cf. (Equation 10).

### 4.3. Non-Degeneracy

The following example shows that even if *d* fails to be well structured, this does not guarantee that $M_{d}\left(y\right)$ is degenerate. Indeed, choosing $d={\left(4,2,1\right)}^{⊤}$, $y=\frac{1}{5}{\left(3,1,1\right)}^{⊤}$, one computes
$$\mathrm{ext}\left(M_{d}\left(y\right)\right)=\frac{1}{20}\left\{\left(\begin{array}{c} 12\\ 4\\ 4 \end{array}\right),\left(\begin{array}{c} 13\\ 4\\ 3 \end{array}\right),\left(\begin{array}{c} 13\\ 5\\ 2 \end{array}\right),\left(\begin{array}{c} 11\\ 7\\ 2 \end{array}\right),\left(\begin{array}{c} 10\\ 7\\ 3 \end{array}\right),\left(\begin{array}{c} 10\\ 6\\ 4 \end{array}\right)\right\}$$
(recall Section 4.1 for details on how to evaluate $E_{d,y}$). In particular, $|\mathrm{ext}(M_{d}\left(y\right))|=6=3!$, although *d* is not well structured (1+2<4). The reason for this is that fact that when partitioning the interval [0,7] into subintervals of length (1,4,2) (which is the re-ordering of *d* such that it matches $\frac{y}{d}$ being non-increasing), there is no way to permute these subintervals such that the two small intervals are contained in the large one.

### 4.4. Index Sets $I_{k}^{\sigma }$

In (Equation 7), we defined the index sets $I_{k}^{\sigma }$. Here, we wish to demonstrate how to easily compute them. Using the same example as in Section 4.1, i.e., $y={\left(4,0,1\right)}^{⊤}$ and $d={\left(4,2,1\right)}^{⊤}$, we again have the thermomajorization curve in Figure 3. This curve changes slope only once (i.e., m=1) and it does so at the input 5; thus, $\Delta_{0}=0$, $\Delta_{1}=5$, and $\Delta_{2}=e^{⊤}d=7$. Let us specify the sets $I_{\sigma }^{1},I_{\sigma }^{2}$ for the permutation σ=(231) from Figure 4.

One finds $I_{1}^{\sigma }=\left\{\sigma \left(1\right),\sigma \left(2\right),\sigma \left(3\right)\right\}=\left\{2,3,1\right\}$ because all three subintervals intersect the interval $\left(\Delta_{0},\Delta_{1}\right)=\left(0,5\right)$ corresponding to the first affine linear segment. Moreover, $I_{2}^{\sigma }=\left\{\sigma \left(3\right)\right\}=\left\{1\right\}$ because $\left(\Delta_{1},\Delta_{2}\right)=\left(5,7\right)$ only intersects the third subinterval (with regard to σ).

### 4.5. Extremal *d*-Stochastic Matrices

We already saw that for $y={\left(4,0,1\right)}^{⊤}$ and $d={\left(4,2,1\right)}^{⊤}$, the extreme point corresponding to σ=(231) is $E_{d,y}\left(\sigma \right)={\left(2,2,1\right)}^{⊤}$. Finding a Gibbs-stochastic matrix that maps *y* to this extreme point via Definition 4 amounts to specifying a permutation that orders $\frac{y}{d}$ non-increasingly, e.g., choose τ=(132) because then $\underline{\tau }\frac{y}{d}={\left(\frac{4}{4},\frac{1}{1},\frac{0}{2}\right)}^{⊤}={\left(1,1,0\right)}^{⊤}$. Now, all the information needed for the definition in (Equation 9) is contained in Figure 4.

This figure allows us to easily build $\underline{\sigma }A_{\sigma \tau }{\underline{\tau }}^{-1}$ because the non-zero entries of this matrix correspond to how much of the interval $\left(\sum_{i=1}^{j-1}d_{\tau (i)},\sum_{i=1}^{j}d_{\tau (i)}\right]$ is overlapped by a given element of the partition $\left\{d_{\sigma (i)}:i=1,\ldots ,n\right\}$ (by slight abuse of notation, we identify $d_{\sigma (i)}$ with the interval of corresponding length). For example, $\left(0,d_{\sigma (1)}\right]$ overlaps with $d_{\tau (1)}$ (covering $\frac{2}{4}$ = half of it) but not with $d_{\tau (2)}$, $d_{\tau (3)}$. This means that the first row of $\underline{\sigma }A_{\sigma \tau }{\underline{\tau }}^{-1}$ is given by ${\left(\frac{1}{2},0,0\right)}^{⊤}$. Similarly, the second row reads ${\left(\frac{1}{4},0,0\right)}^{⊤}$. On the other hand, $d_{\sigma (3)}$ in Figure 4 overlaps with all three sections $d_{\tau (1)}$, $d_{\tau (2)}$, $d_{\tau (3)}$, and the corresponding overlap ratios are given by $\frac{1}{4}$, $\frac{1}{1}$, and $\frac{2}{2}$. Hence, the third row of $\underline{\sigma }A_{\sigma \tau }{\underline{\tau }}^{-1}$ is given by ${\left(\frac{1}{4},1,1\right)}^{⊤}$, which altogether yields
$$\underline{\sigma }A_{\sigma \tau }{\underline{\tau }}^{-1}=\left(\begin{array}{ccc} \frac{1}{2} & 0 & 0\\ \frac{1}{4} & 0 & 0\\ \frac{1}{4} & 1 & 1 \end{array}\right)\;\Leftrightarrow \;A_{\sigma \tau }={\underline{\sigma }}^{-1}\left(\begin{array}{ccc} \frac{1}{2} & 0 & 0\\ \frac{1}{4} & 0 & 0\\ \frac{1}{4} & 1 & 1 \end{array}\right)\underline{\tau }=\left(\begin{array}{ccc} \frac{1}{4} & 1 & 1\\ \frac{1}{2} & 0 & 0\\ \frac{1}{4} & 0 & 0 \end{array}\right)\;.$$
One readily verifies that $A_{\sigma \tau }\in s_{d}\left(3\right)$ maps *y* to $E_{d,y}\left(\sigma \right)={\left(2,2,1\right)}^{⊤}$. Another observation to make here is that ${\left(A_{\sigma \tau }\right)}_{ij}$ is always given by the portion of $d_{j}=d_{\tau (\tau^{-1}\left(j\right))}$ that is covered by $d_{i}=d_{\sigma (\sigma^{-1}\left(i\right))}$. In the above example, ${\left(A_{\sigma \tau }\right)}_{21}=\frac{1}{2}$ because half of $d_{1}=d_{\tau (1)}$ is covered by $d_{2}=d_{\sigma (1)}$ in Figure 4.

There is more to uncover here: on the one hand, the τ that we chose is not the only permutation that orders $\frac{y}{d}$ non-increasingly in this example, and, on the other hand, we saw in Section 4.1 that there exists a permutation $\sigma^{\prime }\in S_{3}$ other than the σ that we chose that is mapped to ${\left(2,2,1\right)}^{⊤}$ under $E_{d,y}$. Thus, we can apply the above procedure to 2·2=4 combinations of permutations (σ,τ), which all yield extremal Gibbs-stochastic matrices mapping *y* to ${\left(2,2,1\right)}^{⊤}$. These are given in Table 2 below.

## 5. Conclusions

In this work, building upon [8,15,17,18,19,20], we further explored thermomajorization in the quasi-classical realm, as well as the rich geometry of the associated polytope. The former notion comes from the resource theory approach to quantum thermodynamics, and, in particular, the corresponding set of allowed operations, known as thermal operations.

Inspired by transportation theory, the core notions of this work were “stable” and “well-structured” Gibbs vectors, which are simple conditions on the spectrum of a Gibbs state. We found that these concepts relate to and give rise to conceptional insights and unexpected results regarding system properties and state transfers. On the one hand, quasi-classical cyclic state transfers with regard to thermomajorization are impossible if and only if the Gibbs vector is stable (which is the generic case). Put differently, within the model of (quasi-classical) thermal operations, performing cyclic state transfers in general comes with a non-zero work cost. On the other hand, we uncovered two connections to the notion of subspaces in equilibrium. (1). Thermal operations can bring any subspace in equilibrium out of equilibrium without having to expend work. This generalizes the intuitive fact that a system in a Gibbs state can be brought out of equilibrium by coupling it to a non-equilibrium system. Note that for the latter scenario—while any out-of-equilibrium system is necessarily a resource—our result shows that this is the only price that one has to pay, i.e., once the systems are coupled, there is some process on the composite system that takes the original system out of the Gibbs state and that can be implemented at no work cost. (2). The existence of subspaces in equilibrium is reflected in the geometry of the thermomajorization polytope. Indeed, assuming the well-structuredness of the Gibbs state—which is equivalent to the system’s temperature exceeding a critical value—the existence of a subspace in equilibrium corresponds to the polytope having degenerate corners.

While we explored the case of quasi-classical states in detail, as usual in quantum thermodynamics, the general case is vastly more difficult. Most notably, the number of extreme points of the thermal operations, as well as the number of extreme points of the future thermal cone, is uncountable for non-classical initial states [7,21]. Hence, one loses access to tools from the theory of convex polytopes. One of the few notions that pertain to general systems is the notion of well-structured and stable Gibbs states; investigating whether these notions encode any properties of general thermal cones (such as, e.g., the impossibility of cyclic processes) could be an interesting topic for future research.

## Figures and Tables

**Figure 1 entropy-26-00106-f001:**
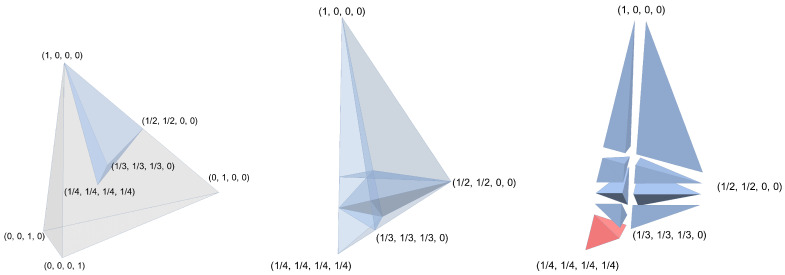
Illustration of stability and well-structuredness. Consider $d\in R_{++}^{4}$, where, without loss of generality, $e^{⊤}d=1$. Then, *d* lies in the relative interior of the standard simplex shown on the left. By reordering its entries in a non-increasing fashion—cf. Remark 1—we can assume that *d* lies in the ordered Weyl chamber shown in the middle. The unstable points are composed of the walls of the Weyl chamber as well as five planes intersecting the Weyl chamber. These planes cut the Weyl chamber into nine subchambers, and the one that includes the maximally mixed state e/4 (highlighted in red on the right) contains exactly the well-structured Gibbs vectors.

**Figure 2 entropy-26-00106-f002:**
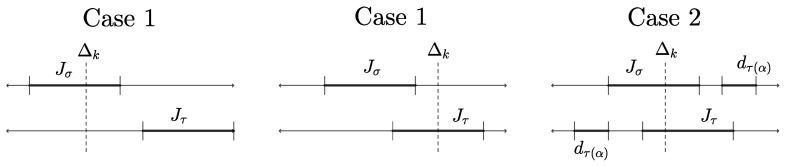
Possible combinations of whether $J_{\sigma }$ and $J_{\tau }$ intersect, and where $\Delta_{k}$ lies relative to $J_{\sigma },J_{\tau }$.

**Figure 3 entropy-26-00106-f003:**
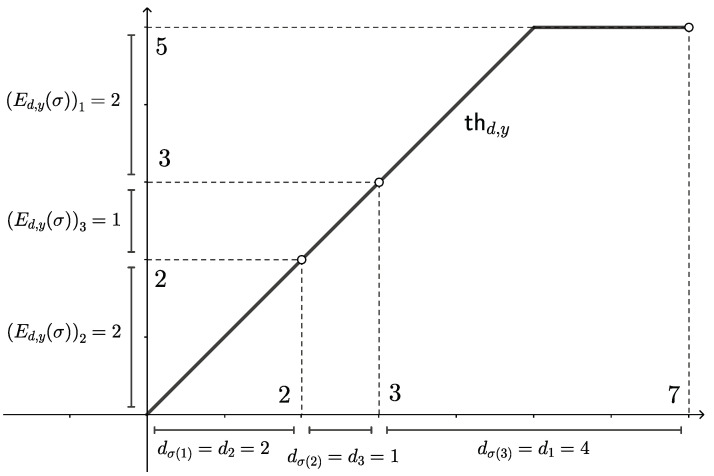
Visualization of the detailed calculation of $E_{d,y}\left(\sigma \right)$ for σ=(231). First, we extract the value of ${\mathrm{th}}_{d,y}$ at $d_{\sigma (1)}=d_{2}=2$, which, because we considered the second entry of *d*, becomes the second entry of $E_{d,y}\left(\sigma \right)$. Next, we add $d_{\sigma (2)}=d_{3}=1$ to the previous *x*-axis value; then, the third entry of $E_{d,y}\left(\sigma \right)$ is the corresponding increment ${\mathrm{th}}_{d,y}\left(3\right)-{\mathrm{th}}_{d,y}\left(2\right)=1$. Finally, we add $d_{\sigma (3)}=d_{1}$ to our position on the *x*-axis to arrive at $e^{⊤}d=7$, so the first entry of $E_{d,y}\left(\sigma \right)$ is given by the final increment ${\mathrm{th}}_{d,y}\left(7\right)-{\mathrm{th}}_{d,y}\left(3\right)=2$.

**Figure 4 entropy-26-00106-f004:**
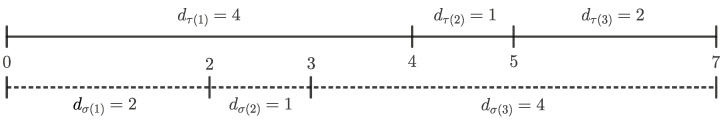
Comparison of the sets $\left\{\sum_{i=1}^{j}d_{\tau (i)}:j=1,\ldots ,n\right\}$ (top) and $\left\{\sum_{i=1}^{j}d_{\sigma (i)}:j=1,\ldots ,n\right\}$ (bottom) as subsets of the real number line.

**Table 1 entropy-26-00106-t001:** Image of $E_{d,y}$ for $y={\left(4,0,1\right)}^{⊤}$ and $d={\left(4,2,1\right)}^{⊤}$ with intermediate steps.

σ	$\;{\mathrm{th}}_{d,y}\left(d_{\sigma (1)}\right)\;$	$\;{\mathrm{th}}_{d,y}\left(d_{\sigma (1)}+d_{\sigma (2)}\right)\;$	$E_{d,y}\left(\sigma \right)$
(1 2 3)	4	min{6, 5} = 5	$\left(\begin{array}{c} 4\\ 5-4\\ 5-5 \end{array}\right)=\left(\begin{array}{c} 4\\ 1\\ 0 \end{array}\right)$
(3 1 2)	1	min{5, 5} = 5	$\left(\begin{array}{c} 5-1\\ 5-5\\ 1 \end{array}\right)=\left(\begin{array}{c} 4\\ 0\\ 1 \end{array}\right)$
(2 3 1)	2	min{3, 5} = 3	$\left(\begin{array}{c} 5-3\\ 2\\ 3-2 \end{array}\right)=\left(\begin{array}{c} 2\\ 2\\ 1 \end{array}\right)$
(2 1 3)	2	min{6, 5} = 5	$\left(\begin{array}{c} 5-2\\ 2\\ 5-5 \end{array}\right)=\left(\begin{array}{c} 3\\ 2\\ 0 \end{array}\right)$
(1 3 2)	4	min{5, 5} = 5	$\left(\begin{array}{c} 4\\ 5-5\\ 5-4 \end{array}\right)=\left(\begin{array}{c} 4\\ 0\\ 1 \end{array}\right)$
(3 2 1)	1	min{3, 5} = 3	$\left(\begin{array}{c} 5-3\\ 3-1\\ 1 \end{array}\right)=\left(\begin{array}{c} 2\\ 2\\ 1 \end{array}\right)$

**Table 2 entropy-26-00106-t002:** Different combinations of $\sigma ,\tau \in S_{3}$ and the corresponding matrix $\underline{\sigma }A_{\sigma \tau }{\underline{\tau }}^{-1}$ (left), as well as $A_{\sigma \tau }$ (right).

σ↓/τ→	(1 3 2)	(3 1 2)	σ↓/τ→	(1 3 2)	(3 1 2)
(2 3 1)	$\left(\begin{array}{ccc} \frac{1}{2} & 0 & 0\\ \frac{1}{4} & 0 & 0\\ \frac{1}{4} & 1 & 1 \end{array}\right)$	$\left(\begin{array}{ccc} 1 & \frac{1}{4} & 0\\ 0 & \frac{1}{4} & 0\\ 0 & \frac{1}{2} & 1 \end{array}\right)$	(2 3 1)	$\left(\begin{array}{ccc} \frac{1}{4} & 1 & 1\\ \frac{1}{2} & 0 & 0\\ \frac{1}{4} & 0 & 0 \end{array}\right)$	$\left(\begin{array}{ccc} \frac{1}{2} & 1 & 0\\ \frac{1}{4} & 0 & 1\\ \frac{1}{4} & 0 & 0 \end{array}\right)$
(3 2 1)	$\left(\begin{array}{ccc} \frac{1}{4} & 0 & 0\\ \frac{1}{2} & 0 & 0\\ \frac{1}{4} & 1 & 1 \end{array}\right)$	$\left(\begin{array}{ccc} 1 & 0 & 0\\ 0 & \frac{1}{2} & 0\\ 0 & \frac{1}{2} & 1 \end{array}\right)$	(3 2 1)	$\left(\begin{array}{ccc} \frac{1}{4} & 1 & 1\\ \frac{1}{2} & 0 & 0\\ \frac{1}{4} & 0 & 0 \end{array}\right)$	$\left(\begin{array}{ccc} \frac{1}{2} & 1 & 0\\ \frac{1}{2} & 0 & 0\\ 0 & 0 & 1 \end{array}\right)$

## Data Availability

No new data were created or analyzed in this study. Data sharing is not applicable to this article.

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
