# Peer review of "The Thermomajorization Polytope and Its Degeneracies"

_entropy, 2024, doi:10.3390/e26020106_

Round 1

Reviewer 1 Report

Comments and Suggestions for Authors

The paper presents a rigorous picture of the structural properties of resource theory of Thermal Operations in a semi-classical regime. This enables characterization of the ability of performing cyclic transformations, and of driving the system out of a relative equilibrium defined within a subspace. In the first case, the generic inability of performing non-trivial global cyclic transformations is associated with the fact that Gibbs states are typically stable. In the other case it is established that every non-equilibrium state with relative equilibrium can be brough out if it by Thermal Operations. Moreover, in high temperature regime (with critical temperature dependent on the system in consideration), limited number of extreme points of the set of states reachable from a given state is a witness of the state’s relative equilibrium. The paper also presents a procedure for determining extreme Gibbs-stochastic maps transforming an initial semi-classical state to extreme points of the set of states reachable by Thermal Operations.

As the structure of Thermal Operations becomes elaborate even in semi-classical regime, to me the contribution of the paper lies in identifing consequences of typical features of the system and environment. We learn that cyclic non-trivial global transformations under Thermal Operations require fine-tuning of the system parameters with respect to the environment, which suggest that operationally, precise global cyclicity should not be attainable. On the other hand, relative equilibrium, while it is associated with simplification of the set of maps required to atttain all allowed states in high temperature regime, is identified as generically unstable under Thermal Operations. Hence, Thermal Operations should generically possess a complex structure, at least in the high temperature limit. I consider these as valuable insights into the typical structure of Thermal Operations.

The paper is written in a clear and well-structured manner, and all the relevant recent publications in the topic are duly cited. The manuscript is scientifically sound, and conclusions are consistent with the evidence and arguments presented. Figures and tables are appropriate.

I have 2 remarks:

1. In line 271, the orthoginality symbol should be related to operator A_p, not identity.

2. Definition 2 of non-equilibrium refers to sets of indices. Hence, I find it confusing when in the text, the authors write about “subsystem in equilibrium” (which suggests some tensor product structure of the system), instead of “system with subspaces in equilibrium”, or something equivalent. This distinction is not of purely linguistic character: when made, it enables to formulate a question about consequences of ‘subspace equilibrium’ induced by preparation of a subsystem in equilibrium (a trivial scenario in Thermal Operation framework), as oposed to consequences of ‘genuine’ subspace equilibrium. I am curious if this distinction has any consequences in the set of reachable states, when catalysis is considered. From the point of view of the structure of the set of Gibbs-preserving matrices, I would expect ‘subsystem non-equilibrium’ to lead to significant reduction in the number of extreme reachable points, and hence in the number of extreme maps.

In summary, in recommend publication of the paper without major changes.

Author Response

We would like to thank the referee for their favorable review and their helpful comments. We agree that the term "subsystem in equilibrium" may be misleading and we followed the referee's suggestion to change it to "subspace in equilibrium" everywhere. Regarding point 1 (orthogonality symbol in line 271) we have re-written and slightly expanded the paragraph in question to hopefully clarify the point we were trying to make there.

Reviewer 2 Report

Comments and Suggestions for Authors

I think that the work provides very interesting new results in the resource theory approach to quantum thermodynamics. Nevertheless, I think that it can seem too abstract even for a quantum thermodynamics specialist using other approaches and working in other subareas (e.g. quantum engines, fluctuation theorems, thermalization, thermodynamics of strongly-coupled open systems, etc.), let alone a broader audience. So I suggest grounding these results a bit in Conclusions by adding a few sentences about potential specific physical applications of the results obtained. I think that it will show the significance of your results to a broad physical audience of Entropy readers.

And I have another minor suggestion. As far as I know, Entropy allows “Abbreviations” section, before the appendices. So I suggest putting there even relatively widespread abbreviations such as “w.l.o.g.” and “w.r.t.”.

Author Response

We would like to thank the referee for their favorable review and their helpful comments. As suggested we have included an "abbreviations" section prior to the appendix. Moreover, as requested, we have expanded the "conclusions" section: We believe that the main takeaways from our article are fundamental insights into elementary physical processes (rather than immediate physical applications) which is now reflected by two additional paragraphs where we attempt to convey these insights---in more general language---to a broader audience.

Reviewer 3 Report

Comments and Suggestions for Authors

The paper is devoted to the resource theory of quantum thermodynamics. Namely, the so called thermal operations are treated as "free". The problem is to describe the set of states that can be achieved from a given initial state by thermal operations. This problem is well-known and gives rise to the resource theory of quantum thermodynamics, which is actively studied now.

In the present paper, the authors establish parallels between the resource theory of thermodynamics and the transportation problem from the optimisation theory. Namely, the authors translate known concepts from the theory of the transportation problem to the language of the resource theory of thermodynamics and introduce the concepts of the stability and well-structuredness of the Gibbs (thermal) state to the latter. They prove that these new concepts can be useful in the resource theory of thermodynamics.

Namely, the authors prove three theorems for the case where the density matrices of quantum states are diagonal in the energy eigenbasis. The first theorem relates the property of stability with the existence of a cyclic thermal process. The second theorems shows that, if only a subsystem of a given system is in the equilibrium, than arbitrary energy levels of the systems can be driven out of the equilibrium by thermal operations. Mainly, this scenario is interesting in the context of catalytic transitions, where an additional system out of equilibrium is attached to the considered system (which can be in equilibrium). The third theorem provides an evidence of either the existence of a subsystem in equilibrium or non-well-structuredness of the Gibbs state.

The results are very interesting and clearly presented. The manuscript is definitely worth to be published.

I cannot find any "technical" drawbacks in the text and have only a general comment which can be optionally consider by the authors. I'm afraid that I don't fully understand the relation of Theorem 2 to the introduced concepts of stability or well-structuredness. It seems that this result stands apart from the other results of the paper and is not related to the introduced notions (again, in contrast to Theorems 1 and 3). I think, it would be desirable to explain the relation of this theorem to the introduced concepts clearer in order for a reader to better understand the general logic of the paper. Yes, a relation between the existence of a subsystem out of equilibrium and well-structuredness appears in Theorem 3, but it seems that it is not directly related to Theorem 2. Theorem 3 gives an evidence that there exists a subsystem not in equilibrium, while Theorem 2 uses the existence of such subsystem as a condition.

Author Response

We would like to thank the referee for their favorable review and their helpful comments. As suggested we re-worked the beginning of section 3.2 in order to clarify to the reader that Theorem 2 on its own is independent from the notions of stability & well-structuredness, but that the connection between these will be made clear in the section in question.